# T-Cell Acute Lymphoblastic Leukemia: Biomarkers and Their Clinical Usefulness

**DOI:** 10.3390/genes12081118

**Published:** 2021-07-23

**Authors:** Valentina Bardelli, Silvia Arniani, Valentina Pierini, Danika Di Giacomo, Tiziana Pierini, Paolo Gorello, Cristina Mecucci, Roberta La Starza

**Affiliations:** 1Hematology and Bone Marrow Transplantation Unit, Laboratory of Molecular Medicine (CREO), Department of Medicine and Surgery, University of Perugia, 06132 Perugia, Italy; valentina.bardelli@studenti.unipg.it (V.B.); silvia.arniani@gmail.com (S.A.); pierinivalentina@yahoo.it (V.P.); danikadigiacomo@libero.it (D.D.G.); tiziana.pierini2005@libero.it (T.P.); gorpaolo@libero.it (P.G.); cristina.mecucci@unipg.it (C.M.); 2Department of Chemistry, Biology and Biotechnology, University of Perugia, 06132 Perugia, Italy

**Keywords:** T-ALL, genomic profile, molecular-cytogenetic markers

## Abstract

T-cell acute lymphoblastic leukemias (T-ALL) are immature lymphoid tumors localizing in the bone marrow, mediastinum, central nervous system, and lymphoid organs. They account for 10–15% of pediatric and about 25% of adult acute lymphoblastic leukemia (ALL) cases. It is a widely heterogeneous disease that is caused by the co-occurrence of multiple genetic abnormalities, which are acquired over time, and once accumulated, lead to full-blown leukemia. Recurrently affected genes deregulate pivotal cell processes, such as cycling (*CDKN1B, RB1, TP53*), signaling transduction (*RAS* pathway, *IL7R/JAK/STAT, PI3K/AKT*), epigenetics (*PRC2* members, *PHF6*), and protein translation (*RPL10, CNOT3*). A remarkable role is played by *NOTCH1* and *CDKN2A*, as they are altered in more than half of the cases. The activation of the *NOTCH1* signaling affects thymocyte specification and development, while *CDKN2A* haploinsufficiency/inactivation, promotes cell cycle progression. Among recurrently involved oncogenes, a major role is exerted by T-cell-specific transcription factors, whose deregulated expression interferes with normal thymocyte development and causes a stage-specific differentiation arrest. Hence, *TAL* and/or *LMO* deregulation is typical of T-ALL with a mature phenotype (sCD3 positive) that of *TLX1, NKX2-1*, or *TLX3*, of cortical T-ALL (CD1a positive); *HOXA* and *MEF2C* are instead over-expressed in subsets of Early T-cell Precursor (ETP; immature phenotype) and early T-ALL. Among immature T-ALL, genomic alterations, that cause *BCL11B* transcriptional deregulation, identify a specific genetic subgroup. Although comprehensive cytogenetic and molecular studies have shed light on the genetic background of T-ALL, biomarkers are not currently adopted in the diagnostic workup of T-ALL, and only a limited number of studies have assessed their clinical implications. In this review, we will focus on recurrent T-ALL abnormalities that define specific leukemogenic pathways and on oncogenes/oncosuppressors that can serve as diagnostic biomarkers. Moreover, we will discuss how the complex genomic profile of T-ALL can be used to address and test innovative/targeted therapeutic options.

## 1. Introduction

T-cell acute lymphoblastic leukemia (T-ALL) represents 15% of pediatric and 25% of adult ALL cases. Historically considered an aggressive subtype of ALL, new pediatric-oriented protocols, in adults, and intensified treatments, based on the persistence of minimal residual disease (MRD) in both children and adults, have greatly improved patient outcomes [1]. The genetic background of T-ALL is widely heterogeneous as a huge number of oncogenes and oncosuppressors, often intersecting on a few pivotal cellular processes, have been associated with the onset and/or progression of the leukemia. Furthermore, alternative molecular mechanisms can cause gene deregulation (Figure 1). Hence, the molecular cytogenetic diagnosis of T-ALL is not currently adopted, as a unifying test able to pick up the large spectrum of the involved genes and the alternative molecular mechanisms underlying their deregulation, is not available for clinical purposes. Therefore, although our knowledge on the genetic background of T-ALL has been greatly elucidated in the last decade, and specific leukemogenic pathways have been identified, the clinical impact of this bio-molecular information is still largely unknown. The majority of available studies focused on individual markers, sometimes yielding misleading results, as they overlooked the complexity and heterogeneity of the disease. On the other hand, when more than one T-ALL associated abnormality is taken into account, a strong predictive value of the leukemia genomic profile emerged. If it is combined with MRD determination, a significant improvement of risk assessment in both pediatric and adult patients has been obtained [2,3].

This review will focus on the most relevant biological aspects of T-ALL, describing how these leukemias can be classified into relevant genetic entities closely related to the stage of immunophenotypic T-cell differentiation. In addition, we will report on the incidence and distribution of recurrent abnormalities that can be regarded as putative prognostic and/or actionable targets.

## 2. Driver Oncogenic Events: Classification into Type A and Type B Abnormalities

According to their reciprocal distribution and effects of their deregulation, T-ALL-related abnormalities can be distinguished as type A and type B [4]. Type A abnormalities are mutually exclusive and cause the up-regulation of (onco)genes coding for transcription factors critical in hematopoiesis and/or T-cell development, maturation, and differentiation. These genetic events mark distinct subgroups, i.e., *TAL/LMO*, *HOXA*, *TLX3*, *TLX1*, *NKX2-1/2-2*, *MEF2C*, allowing the molecular classification of at least 70% of T-ALL cases, with different distribution in children and adults (Figure 2) [5,6,7,8,9]. Type B abnormalities, on the other hand, involve genes that code for different protein families, such as epigenetic factors, tyrosine kinases, ribosomal proteins, and proteins that belong to signaling pathways. They are distributed across all genetic subgroups, albeit not randomly, as privileged associations occur between primary and secondary genetic changes. Thus, concurrent primary and secondary events trace unique leukemogenic pathways that rely on specific cooperative activities.

## 3. Mature T-ALL and the *TAL/LMO* Transcriptional Complex

First identified by Ferrando A et al. [5], *TAL1* overexpression characterizes 30–45% of pediatric and 10–15% of adult T-ALL (Figure 2). These T-ALL harbor chromosomal rearrangements, or alternatively noncoding sequence mutations, which cause the activation of members of the basic helix-loop-helix (bHLH) family of transcription factors (*TAL1* and *TAL2*) and/or of the LIM-only domain (*LMO1*, *LMO2*, and *LMO3*) proteins [8,11,12,13]. TAL and LMO proteins belong to the same transcriptional complex and are frequently co-deregulated in T-ALL, suggesting a cooperative role in T-leukemogenesis. Consistent with this model, the oncogenic activity of *LMO1* or *LMO2* is markedly enhanced by *TAL1,* in transgenic mice [14,15]. *TAL* and *LMO* leukemogenic activity, demonstrated in in vitro and in vivo studies, was the consequence of the altered expression of their downstream targets, such as *TRIB2*, *NKX3.1, RUNX1*, *GATA3*, and *MYB* [16,17].

Due to the aforementioned characteristics, T-ALL with abnormalities of *TAL* and/or *LMO* genes (Figure 3), are typically grouped together into the *TAL/LMO* category as they share the same gene expression signature. Their immunophenotype mostly resemble mature T-cell development arrest (sCD3 positive), and while showing a low incidence of *NOTCH1* mutations (~40% of cases) (Figure 4), have a privileged association with *PTEN* inactivation, 6q14-q16 deletions, and *MYC* translocations, all behaving as poor prognostic markers [8,18,19].

*MYC* translocations occur in approximately 5% of pediatric and adult T-ALL, involve both *TR* and non-*TR* partners, and invariably cause *MYC* over-expression [8,18]. They mark a subgroup of *NOTCH1*-independent T-ALL, which are characterized by a high incidence of *PTEN* deletions/loss-of-function mutations. Preliminary studies suggested that *MYC* translocations were high risk markers, since they have been associated with high white blood cell count, poor response to glucocorticoid pretreatment, and poor response to standard chemotherapy [18,19,20]. Interestingly, the therapeutic sensitivity of *MYC*-positive T-ALL towards BET bromodomain inhibitors has been observed in preclinical models [19].

Although widely heterogeneous, 6q14-q16 deletions encompass the oncosuppressor *CASP8AP2*/6q15 in ~90% of cases, where it is significantly downregulated [21]. Particularly, deletion of *CASP8AP2*/6q15 occurs in ~20% of *TAL/LMO* T-ALL and predicts poor response to therapy with the persistence of minimal residual disease (MRD) [8,21]. Besides *CASP8AP2*, two further candidate tumor suppressor genes, i.e., *SYNCRIP* (encoding hnRNP-Q) and *SNHG5* (that hosts snoRNAs), have been recently identified at 6q14.3, ~4Mb apart from *CASP8AP2*. When codeleted/haploinsufficient, *SYNCRIP* and *SNHG5,* alter ribosomal functions, increase leukemia-initiating cell activity, and induce tumor progression [22]. The size of the 6q deletions and the involvement of several putative tumor suppressors suggest that the haploinsufficiency of more than one gene may be at the basis of these genomic losses. Furthermore, diverse types of 6q deletions might be related to distinct genetic subgroups.

## 4. Early/Immature T-ALL: Transcriptome and *HOXA*, *MEF2C*, and *BCL11B* Oncogene Deregulation

Early and immature T-ALL are two distinct subtypes of T-cell differentiation arrest of leukemia. Early T-ALL, also known as pre-T ALL, were first defined by the EGIL group [23], by the expression of the cCD3 T-cell lymphoid markers with/without CD2, CD5, and CD8, but negative for CD1a and sCD3. Even more immature is the so-called early T-cell precursor (ETP) ALL, which was first reported by Coustan-Smith in 2009, and that largely overlaps with the pro-T subtype of the EGIL classification [23,24]. ETP-ALL are recognized by a distinct immature phenotype, characterized by the expression of at least one stem cell (CD34, CD117) and/or myeloid markers (CD13, CD33, HLA-DR, CD11b, CD65) in at least 25% of leukemic cells, negativity or weak expression (<75%) of CD5, and no expression of CD4 and CD8 [24]. It is worth mentioning that, similar to ETP ALL is a subset of immature T-ALL, called near-ETP, which despite having overlapping characteristics, exhibits CD5 antigen expression at levels that are not low enough to meet ETP-ALL criteria [9,24].

It has been demonstrated that the state of maturation arrest of T-cell leukemias, determines the specific anti-apoptotic protein on which depends cell survival [25]. In ETP-ALL, leukemic blasts depend on the anti-apoptotic protein *BCL2*, which is widely over-expressed in this subtype of T-ALL [5,25]. Based on these data, preclinical in vitro studies have exploited *BCL2* as a putative target in immature T-ALL, using the specific inhibitor BH3 mimetic ABT-199, also known as venetoclax [26]. ABT-199 showed a strong anti-leukemic activity and a synergistic effect when combined with chemotherapy or steroids in immature T-ALL but not in more mature T-ALL subtypes [26]. A remarkable result in the use of venetoclax in patients with immature T-ALL, has been obtained by our group, in collaboration with Dr. G Roti, University of Parma. Combining molecular-cytogenetic and Drug Sensitivity and Resistance Profiling (DSRP) studies, we confirmed *BCL2* over-expression in three cases of immature T-ALL, and used this marker to prioritize the selection of the most effective compounds, as identified by DSRP. Accordingly, these patients, all with refractory/early relapsed ETP/near ETP ALL, were treated with ABT-199 combined with bortezomib, the latter used to overcome known resistance linked to ABT-199 administration, and obtained complete/partial hematological remission. Subsequently, two underwent hematopoietic stem cell transplantation, and achieved a stable cytogenetic remission [27].

ETP-ALL have a distinctive gene expression signature characterized by high levels of expression of myeloid markers, i.e., *LYL1, LMO2, CD34, CCND2, KIT, GATA2*, and *CEBPA*, and low levels of expression of genes related to thymocyte differentiation, such as *CD1, CD4, CD8, TCF7*, and *LEF1* [5,6,24]. Although ETP ALL cases share a specific signature, their genomic profile is extremely heterogeneous, with alterations of many different genes. They include activating mutations of genes encoding for cytokine receptors and RAS signaling (*NRAS, KRAS, FLT3, IL7R, JAK3, JAK1, SH2B3*, and *BRAF*), deletions/loss-of-function mutations of genes involved in hematopoietic development (*GATA3, ETV6, RUNX1, IKZF1*, and *EP300*), and abnormalities of histone-modifying genes (*EZH2, EED, SUZ12*, and *SETD2*). On the other hand, abnormalities of T-ALL-related genes, such as *NOTCH1* and *CDKN2AB*, are rare (Figure 4) [28,29]. Despite the genomic heterogeneity, activation of *HOXA*, *MEF2C*, or *BCL11B*, characterize specific subtypes of immature cases, accounting for approximately half of ETP-ALL (Figure 5).

### 4.1. HOXA Cluster Genes

The first evidence of an oncogenic role of *HOXA* genes in T-ALL pathogenesis was highlighted by gene expression profile through the identification of a distinct subgroup of T-ALL with high *HOXA* expression [6]. *HOXA* positive T-ALL harbored a wide spectrum of different genomic abnormalities affecting *HOXA* itself, *KMT2A*, or *MLLT10*, indicating different mechanisms of *cis* or *trans* activation of the *HOXA* cluster genes can occur [6,30,31]. Further enriching the *HOXA* T-ALL subgroup, there are abnormalities of two promiscuous shuttling nucleoporins, i.e., *NUP98* or *NUP214*, which represent additional mechanisms of *HOXA* activation in both pediatric and adult T-ALL (Figure 6) [32,33,34]. Although usually mutually exclusive, concomitant rearrangements of the *HOXA* and *MLLT10* genes have been reported in rare cases of T-ALL [8,35].

*HOXA*-positive T-ALL have an immature, ETP, or near ETP phenotype, in 40–45% of cases [8,36]. The poor overall survival of *HOXA* positive T-ALL patients, as a whole, as well as the unfavorable predictive/prognostic impact of genomic abnormalities underlying *HOXA* up-regulation, have been reported by several groups [36,37,38,39]. However, *HOXA* related abnormalities do not appear to predict a poor outcome when occurring in non-ETP-ALL cases [36,38,40].

### 4.2. MEF2C Expression

In 2011, Homminga et al. [7] described an immature cluster of T-ALL that included cases with *HOXA*-activating abnormalities and cases lacking known driving oncogenic events. About 30% of these cases was characterized by high levels of expression of *MEF2C*, which was supposed to be driven by rearrangements of *MEF2C*, or of transcription factors that target *MEF2C* (*NKX2.5*, *PU.1*), or *MEF2C*-associated cofactors (*NCOA2*) [7]. Functional experiments conducted in T-ALL cell lines supported the role of *MEF2C* as a transcriptional regulator of genes that are highly expressed in immature T-ALL cases, such as *LMO2*, *LYL1*, and *HHEX* [7]. Moreover, they also showed that *MEF2C* causes a T-cell differentiation block at the immature stage of differentiation [7]. It is however worth mentioning that high levels of *MEF2C* expression can also be found in non-ETP-ALL [41,42].

### 4.3. The BCL11B-a Entity

*BCL11B*, a transcription factor essential for the early T-cell lineage commitment, is typically expressed in thymocytes, starting from the transition between DN2a and DN2b stage. In human T-ALL, it undergoes translocations alternatively activating *TLX3* or *NKX2-5*, but it may also act as a tumor suppressor gene undergoing loss-of-function mutations and/or deletions in 13% and 3% of T-ALL and ETP-ALL, respectively.

A new leukemia subtype (*BCL11B*-a), with a variable phenotype, characterized by coexpression of myeloid and T-lymphoid markers, and ranging from ETP-ALL to immature AML, has been recently identified by our group in adults and by Montefiori LE et al. in children [10,43]. This immature form of leukemia is driven by transcriptional activation of *BCL11B*, due to chromosomal rearrangements that juxtapose *BCL11B* to diverse, active super-enhancers, or to the ZEB2 gene, generating a *ZEB2-BCL11B* fusion. These rearrangements differ from the other T-ALL translocations involving the 14q32 region, such as the t(5;14)(q35;q32)/*BCL11B-TLX3* (see above), due to a more centromeric cluster of 14q32 breakpoints that can be reliably identified with a specifically designed fluorescence in situ hybridization assay. An alternative mechanism of *BCL11B* activation is represented by a focal 2.5Kb amplification that generates a super-enhancer from a noncoding element distal to the gene [43]. In vitro studies have shown that *BCL11B*, even in the absence of Notch signaling, is sufficient to drive a T lineage expression program in hematopoietic progenitor cells through the activation of genes characteristic of T-cell differentiation, block of the myeloid differentiation, and expansion of a subpopulation with a T-lineage immunophenotype.

*BCL11B*/14q32 rearrangements/over-expression behave as disease biomarkers present at diagnosis and relapse, but not at remission. They typically cooccur with *FLT3* mutations and mutations of epigenetic modulators, most frequently *DNMT3A, TET2*, and/or *WT1* gene. Furthermore, *BCL11B*-a AL have a distinct expression profile, characterized by deregulation of *BCL11B* target genes, inhibition of the T-cell differentiation program, and activation of the *JAK/STAT*. As predicted by the genomic profile, *BCL11B*-a AL cases are sensitive to tyrosine kinase and *JAK/STAT* inhibitors, i.e., NVP-BVB808, Momelotinib, Fedratinib, and NVP-BSK805 [10]

## 5. Cortical CD1a Positive T-ALL Are Characterized by *TLX1*, *NKX2-1*, or *TLX3* Over-Expression

Common features of this subgroup of T-ALL are CD1a expression, differentiation arrest at the cortical (DN3-DP) stage of T-cell development, and rearrangement/over-expression of *TLX1* (in adults) or *NKX2.1* (in children) [5,7]. The latter subgroup includes pediatric cases harboring rearrangements of *NKX2.1*, *NKX2.2*, or *MYB* [7]. Transcriptome studies have shown that these cases are characterized by the altered cell cycle, DNA replication, and spindle assembly [5,7]. While mitotic defects represent a direct effect of *TLX1* over-expression, which drives aneuploidy in the earliest stage of leukemogenesis, altered cell cycle might be related to deletion of *CDKN2AB* (85%) and/or *CDKN1B* (25%) cell cycle regulators [8]. These leukemia subtype has a high prevalence of *NOTCH1* mutations (Figure 4) [9]. Other recurrent secondary events are deletions/mutations of *BCL11B* (12% vs. 3% in the overall population), herein behaving as a putative oncosuppressor, and mono-/bi-allelic deletions of *PTPN2*, a ubiquitous nontransmembrane tyrosine phosphatase, found in 20% of *TLX1/NKX2.1* positive cases [8,9,44]. In addition, pediatric *NKX2.1*-positive cases have a high rate of deletions of *LEF1*, a DNA-binding transcription factor, which interacts with nuclear β-catenin in the WNT signaling pathway, and of mutations of *RPL10*, a ribosomal protein belonging to the 60S ribosomal subunit [8,9,45,46]. Amongst T-ALL, *TLX1/NKX2.1* positive cases have been associated with the best treatment outcome, in both children and adults [47,48,49,50].

It is worth mentioning that, besides *TLX1/NKX2.1* rearrangements, which are virtually always associated with a cortical phenotype, also abnormalities of *TAL/LMO*, *HOXA*, and *TLX3* can be detected in T-ALL, at this stage of differentiation. Notably, ~40% of *TLX3*-positive cases display a cortical phenotype, although this oncogene may also be involved in immature, early, and mature T-ALL [8]. *TLX3* rearrangements mainly occur as a result of t(5;14)(q35;q32) translocation that places the gene under the control of a strong regulatory sequence downstream *BCL11B* at 14q32. In sporadic cases, rearrangement of *TLX3* with *TRB*, *TRAD*, *CDK6*, or unknown partners, have been reported [8,51]. *TLX1* and *TLX3* are closely related genes that belong to the *NKL* family of homeobox transcription factors and regulate the transcription of a widely overlapping set of genes. Accordingly, gene expression-based hierarchical and genomic studies have shown that *TLX1* and *TLX3* positive T-ALL share common oncogenic pathways, although they also have distinguishing features, and a different clinical outcome [5,7,52,53]. In children, high levels of *TLX3* expression, as well as *TLX3* cytogenetic abnormalities, have been associated with both poor and good prognosis, or have not been related to disease outcome [37,49,52,54]. These conflicting data may reflect bias in patient selection, especially with respect to the number of immature and cortical cases included in the study cohorts, and differences in drug sensitivity of this specific T-ALL subtype [55].

## 6. Actionable Deregulated Pathways and Cellular Processes

### 6.1. NOTCH Pathway Activation

The Notch signaling is essential for T-cell lineage development and survival, and for the proliferation of committed T-cell progenitors. In T-ALL, *NOTCH1* is a driving oncogene, whose gain-of-function mutations induce the development of pre-T-cells to leukemia [56]. The first evidence of *NOTCH1* involvement in T-ALL has been provided by the identification of a rare t(7;9)(q34;q34) translocation that involves *TRB* and *NOTCH1*, cell lines, and primary samples [57]. Later on, another rare *NOTCH1* translocation, the t(9;14)(q34;q11) that juxtaposes the gene to the *TRAD* locus, has been described [58,59]. Both translocations cause the transcriptional activation of *NOTCH1* by the *TR* enhancer regions. Nevertheless, the activation of the *NOTCH* signaling is largely dependent on gain-of-function mutations that typically occur at the HD and PEST domains [60]. In addition to *NOTCH1* alterations, the NOTCH pathway can be activated by loss-of-function mutations/deletions of *FBXW7*, encoding for a ubiquitin protein that promotes and mediates the degradation of *NOTCH1* [61,62]. *NOTCH1* and/or *FBXW7* mutations can occur jointly, and are unequally distributed amongst the main genetic subgroup (Figure 4) [61,62,63]. Indeed, present in over 70% of *TLX1*, *TLX3*, and *NKX2.1* positive T-ALL cases, their incidence decreases significantly in both immature and mature T-ALL (≤40% of cases) [8,28,29].

*NOTCH1* mutations have been regarded as both prognostic and predictive biomarkers. However, the prognostic impact of *NOTCH1* mutations, although extensively studied, did not reach unambiguous conclusions, as no good, or even unfavorable, outcomes have been reported for *NOTCH1*-positive T-ALL [64,65,66,67]. These discordant findings can be explained by the variable genomic background, accompanying *NOTCH* activation, which can differently influence the response to therapy. It has also been suggested that differences in therapy intensification may influence the prognostic impact of *NOTCH1/FBXW7* positive T-ALL [39].

While *NOTCH1/FBXW7* cannot be used alone as prognostic markers, there is a general agreement in considering the *NOTCH* signaling as a good candidate for tailored therapy. Preclinical studies have shown a strong therapeutic effect of different *NOTCH1* inhibitors, such as γ-Secretase (GS) and ADAM (Disintegrin Metalloproteases) inhibitors, monoclonal antibody, and molecules that block the activity of the NOTCH1 transcription factor complex, in both T-ALL cell lines and primary samples [68]. In addition, therapeutic targeting of the E3 ubiquitin ligase SKP2, whose expression is regulated by *NOTCH* signaling, has been proven to block T-ALL proliferation in in vitro and in vivo models [69]. Lastly, the sarco-endoplasmic reticulum Ca^2+^-ATPase (*SERCA*) inhibition appeared a valuable way to suppress oncogenic *NOTCH1* signaling, although the severe side effects observed in mouse models precluded their use in humans. Interestingly, promising results with a new *SERCA* inhibitor, the CAD204520, have been recently reported, since CAD204520 exerts a significant anti-leukemic activity in in vitro and xenograft models of *NOTCH*-dependent T-ALL, with very reduced/no off-target toxicity [70].

Although acting as leukemogenic driver events, *NOTCH1* mutations can be primary “initiating events”, even arising prenatally [71], or “late events”, detected in cellular subclones that are lost during disease progression/relapse [72,73,74,75]. These results warn about the use of *NOTCH1* as a marker for MRD determination and also to select patients for testing *NOTCH1* inhibitors, especially when considering these treatments in refractory/relapsed patients. [76].

### 6.2. IL7R/JAK/STAT Signaling Activation

The Interleukin 7 receptor (IL-7R) signaling is required for the commitment, proliferation, and survival of early T-cell progenitors [77]. IL7R activation, through its ligand IL7, induces reciprocal JAK1 and JAK3 phosphorylation, and subsequent recruitment and activation of the transcription regulator STAT5. Once phosphorylated, STAT5 homodimerizes and migrates into the nucleus, acting as a transcription factor [77,78].

About 10% of T-ALL, shows gain-of-function mutations in *IL7R* that drive the constitutive activation of the *JAK/STAT* signaling [79]. Activation of *JAK/STAT* signaling can also be achieved by *JAK3, JAK1,* or *STAT5B* mutations, reported in approximately 7%, 4%, and 1% of T-ALL cases, respectively. Finally, inactivation of *PTPN2* or *DNM*2, both negative regulators of this cascade, can be detected in 10–20% of T-ALL cases [8,80,81]. Overall, abnormalities underlying the *JAK/STAT* activation, are predominantly associated with *TLX1*, *TLX3*, and *HOXA* subgroups, and with the ETP subtype [9,28,79,82,83,84].

Due to its recurrent involvement, the *JAK/STAT* pathway has been regarded as a putative actionable target. In murine xenograft models of ETP-ALL with aberrant activation of this signaling, the *JAK1/2* inhibitor ruxolitinib, was shown to be highly effective [85]. Furthermore, the combination of dexamethasone and ruxolinitib seems to overcome IL7-induced glucocorticoid resistance in T-ALL samples [86].

Being one of the effectors of the *JAK/STAT* pathway, the treonin/serin kinase *PIM1* emerged as a putative downstream hit, in T-ALL with *JAK/STAT* activation. It is worth noting that high levels of *PIM1* expression are typically found in T-ALL harboring activating mutations of *IL7RA, JAK1, JAK3*, and *STAT5B*, and/or loss of function of *PTPN2*. Accordingly, the highest levels of *PIM1* expression have been observed in T-ALL of the *TLX1*, *TLX3*, and *HOXA* subgroups [87,88]. Furthermore, *PIM1* is expressed at high levels, in T-ALL/LBL with a rare t(6;7)(p21;q34) translocation, which by moving the regulatory sequences of *TRB* close to *PIM1*, causes its transcriptional activation [87,88]. Preclinical in vitro and in vivo studies demonstrated that *TRB-PIM1*, as well as *IL7R* positive T-ALL/LBL, benefit from treatments with one of the two pan-PIM inhibitors AZD1208 and TP3654, combined with glucocorticoids or chemotherapy. Overall, these data point to *PIM1,* as a valid actionable hit, in approximately 30% of T-ALL/T-LBL [89,90].

### 6.3. The ABL1/Src-Family Kinases and the Rationale for the Use of Tyrosine-Kinase Inhibitors

Graux et al., for the first time, demonstrated that 6% of pediatric T-ALL express the constitutively phosphorylate tyrosine kinase *NUP214-ABL1*. This rearrangement originates from the fusion and amplification of *NUP214* and *ABL1* in the extrachromosomal structure called episomes. This abnormality is a late event that often cooccurs with *CDKN2A* deletion and/or *NOTCH1* mutations and is typically associated with *TLX1* and *TLX3* subgroups [9,91,92].

T-ALL harboring *NUP214-ABL1* are sensitive to selective tyrosine-kinase inhibitors (TKi), such as imatinib and dasatinib. Interestingly, another imatinib sensitive *ABL1* fusion, the *EML1-ABL1*, has been reported in a single case of T-ALL with a cryptic t(9;14)(q34;q32) [93]. Moreover, it has been recently reported that *ABL2, PDGFRA*, and *PDGFRB*, other *ABL*-class tyrosin-kinases are also recurrently involved in T-ALL, and that they all predict sensitivity towards TKi [94].

The functional consequences of *ABL*-class protein constitutive activation have been referred to the activity of *LCK*, a protein of the SRC family, which is required for the proliferation of *NUP214-ABL1* transformed cells, as demonstrated by in vitro and in vivo studies [95,96]. Furthermore, T-ALL dependency from *LCK* goes beyond the presence of rearrangements of *ABL*-class proteins, as recently shown in pediatric ETP-ALL patients [97]. High levels of *LCK* expression have been related to poor response to glucocorticoid pretreatment, which can, however, be reverted by *LCK* inhibitors, such as dasatinib, bosutinib, nintedanib, and WH-4-023 [98]. Thus, the use of *LCK* inhibitors could be exploited in approximately 30% of T-ALL patients, regardless of the presence of the typical *ABL*-class fusions [99].

### 6.4. PI3K/AKT/mTOR Signaling Predicts Glucocorticoid Resistance

Another oncogenic pathway recurrently activated in T-ALL is the *PI3K/AKT/mTOR*. The activation of this signaling is mainly due to abnormalities/mutations of genes encoding for *PI3K* members, *AKT*, and/or *PTEN*. Deletions and/or loss-of-function mutations of *PTEN*, the main negative regulator of the *PI3K/AK/mTOR* pathway, were first described in 2007 [100], and linked to resistance to *NOTCH1* inhibition in T-ALL cell lines [101]. An additional mechanism of *PTEN* inactivation has been recently found in about 1% of T-ALL, where a 200Kb-1.4Mb deletion, downstream *PTEN*, abrogates the activity of a highly conserved enhancer that modulates *PTEN* transcription [102]. Overall, *PTEN* haploinsufficiency/inactivation occurs in 10–15% of cases [8,102,103], is almost exclusively associated with abnormalities of the *TAL/LMO* subgroup (~20% of cases), and rarely cooccur with *NOTCH1/FBXW7* mutations [104]. While *PTEN* mutations have not been associated with the prognosis, deletions appear to predict lower event-free and overall survival [21,105,106].

Although less frequently, the *PI3K/AKT/mTOR* pathway can be activated by alterations of members of the *PI3K* complex, such as *PIK3CD* and *PIK3R1*, or of the downstream effector *AKT1* (overall about 2% of cases) [9,107]. A remarkable finding has been recently observed in a pediatric patient with Short syndrome, who developed T-ALL at the age of 13 [108]. The patient had an atypical germline variant of *PIK3R1* [NM_181523.3:c.1457C>T, p.(Ala486Val)] in the iSH2 domain, which is a rare site of germline mutations, in patients with Short syndrome. Still, it is the typical hot-spot region of acquired *PIK3R1* mutations in pediatric T-ALL [9]. Strengthening the close cooperation between this pathway and the *TAL/LMO* transcriptional complex, the leukemic blasts acquired an additional *PTEN* deletion and a *STIL-TAL1* fusion [108]. This clinical observation suggests that SHORT syndrome, due to variants in the iSH2 domain of *PIK3R1*, might be a leukemia predisposing syndrome.

Any alteration causing the activation of the *PI3K/AKT* signaling, has been linked to glucocorticoid resistance. At the same time, preclinical models have shown a synergistic cytotoxic effect of *PI3K/AKT* inhibitors and steroids. For example, combined treatment with AS605240, a selective *PI3Kγ* inhibitor, and steroids resulted in a synergist cytotoxic effect in several T-ALL cell lines, and prevented leukemia progression in PDX T-ALL models [109]. Collectively, these data suggest that T-ALL with any genomic abnormality resulting in activation of the *PI3K/AKT* pathway, can benefit from *PI3K* targeted therapy [109,110].

### 6.5. RAS/MAPK Activation

Abnormalities that activate the *RAS/MAPK* pathway are extremely frequent in pediatric T-ALL at relapse, being detected in approximately 40% of cases, whereas their incidence at diagnosis is much lower (15%) [9,111,112]. Overall, they include hot-spot mutations of *KRAS/NRAS* or *BRAF* [113], gain-of-function mutations of the upstream tyrosin-kinase receptor *FLT3*, or abnormalities of the two negative regulators *NF1* and *PTPN11* [113,114]. The activation of the *RAS/MAPK* signaling has been related to chemoresistance [115,116]. Namely, activation of the *RAS/MAPK* signaling predicts resistance to steroids and methotrexate [112]. These effects can be overtaken by *MEK1/2* inhibitors, such as selumetinib and trametinib that after validation in preclinical studies, have been approved for a phase 2 clinical trial in ALL [111,117]. Abnormalities of the *RAS/MAPK* signaling display a strong enrichment in ETP-ALL and *HOXA* subgroups, but also occur in *TLX1*, *TLX3*, *NKX2.1*, and *NKX2.2* positive T-ALL, although at a lesser extent, while they are absent in the *TAL/LMO* subgroup [9].

### 6.6. Epigenetics in T-ALL

The epigenetic regulators most frequently reported to be involved in T-ALL, are *PHF6*, *KDM6A*, and the members of the Polymcomb repressor complex 2 (PRC2), i.e., *EED*, *EZH2*, and *SUZ12*.

*PHF6* maps at Xq26 undergo deletions and/or loss-of-function mutations in about 15% of pediatric and 35–40% of adult T-ALL. They are almost exclusively found in male patients and are significantly enriched within the *TLX1* and *TLX3* subgroups [118]. *PHF6* inactivation is often associated with genetic lesions of the *JAK/STAT* members, i.e., *IL7R, JAK1, JAK3*, and/or *STAT5B,* suggesting a close cooperation in the leukemogenic process [83]. Germline *PHF6* mutations cause the Börjeson–Forssman–Lehmann syndrome (BFLS) [119], a hereditary X-linked disorder presenting with mental retardation and physical deformities. It has been suggested that BFLS may represent a cancer-predisposing syndrome as one patient with BFLS developed a T-ALL at the age of 9 [120].

The role of *PHF6* in chromatin remodeling and transcriptional regulation has been indicated by its interaction with the chromatin remodeling complex nucleosome remodeling and deacetylase (NurD) and with multiple subunits of the PAF1 transcriptional elongation machinery [121,122,123]. It has been demonstrated that loss of *PHF6* is an early mutational event in leukemia transformation and that Phf6 inactivation enhances hematopoietic stem cell (HSC) long-term self-renewal, and hematopoietic recovery after chemotherapy, thus rendering Phf6 knockout HSCs more quiescent and less prone to stress-induced activation [124]. Further supporting its leukemia-initiating tumor suppressor role, inactivating *Phf6* in hematopoietic progenitors, favors the development of *NOTCH1*-induced T-ALL [124].

*KDM6A* (also known as *UTX*) is an H3K27me3 histone demethylase that functions as a tumor suppressor gene, targeted by loss-of-function mutations in approximately 5–15% of T-ALLs [125,126,127]. *KDM6A* mutations are gender-restricted variants as they have been exclusively found in male patients. It is worth noting that, in females, *KDM6A* escapes chromosome X-inactivation suggesting that females are protected against single copy loss of the gene. Thus, inactivation of one *KDM6A* allele in males contributes to tumor development, while in females, it does not, because cells still express the second wild type allele [127]. Interestingly, *KDM6A*, as well as *PHF6*, is an X-linked tumor suppressor gene that might partially explain the skewed gender distribution in T-ALL toward males on a genetic level.

*EZH2, EED*, and *SUZ12* belong to the *PRC2* (polycomb repressive complex 2) complex, which is an epigenetic modulator that mediates the methylation of *H3K27*. Reduced or abolished *PRC2* activity leads to reduction of *H3K27* methylation, with transcriptional activation of targets, and also induces a mitochondrial-mediated apoptosis resistance. It is interesting to note that a recent study on T-ALL, reported an association between *PRC2* deficiency and a hyper-methylation profile [128].

Overall, mutations or deletions of at least one of the three *PRC2* members take place in 25–30% of T-ALL cases, particularly in immature T-ALL, and have been associated with poor response to chemotherapy [28,129,130]. Similar to *PHF6*, a significant association between *Suz12* inactivation and *JAK3* mutations was found in primary T-ALL [131]. In line with these findings, the inactivation of *Suz12* cooperates with mutant *JAK3* to drive T-cell transformation, and T-ALL development [131]. Interestingly, drug screening has identified inhibitors of the *PI3K/AKT/mTOR* and Vascular endothelial growth factor receptor signaling, and of histone deacetylases, as the most effective compounds towards T-cell lymphoblasts that harbor both *SUZ12* and *JAK3* abnormalities [131].

## 7. Infant T-ALL

T-ALL is very rare in infants (≤12 months), where it has been associated with dismal outcomes. Although still poorly defined, infant T-ALL (iT-ALL) appears to have distinctive genomic and transcriptome profiles with respect to pediatric and adult cases [132,133]. Alterations typically observed in T-ALL, such as *NOTCH1* or *FBXW7* mutations, and *CDKN2AB* deletion, are present at a lower frequency than in adults and children, while new markers, such as the complete deletion of *MLF1*, appear to be specific and recurrent in iT-ALL [132]. Notably, the t(6;7)(q23;q34)/*TRB-MYB*, a very rare translocation in pediatric T-ALL (<3% of cases), was reported in three female patients with iT-ALL, displaying high white blood cell count, central nervous system involvement, and refractory disease or late relapse [134,135,136]. In addition, transcriptome and miRNome sequencing showed a clear separation between infant and pediatric T-ALL with 760 differentially expressed mRNAs and 58 differentially expressed miRNAs. Among others, a significant upregulation of *BRD3*, *KIT*, *BLK*, *FLT1*, *NTRK1*, and *ERBB4*, for which clinically approved compounds were under investigation, has been detected [133]. As observed in B-cell ALL, an amount of evidence indicate that some of the genetic abnormalities of iT-ALL can be acquired during pregnancy [132].

## 8. Therapeutic Opportunity to Be Exploited in T-ALL

While still challenging, we expect great improvement in the diagnosis of T-ALL in the near future. To translate biomolecular information in the clinical practice, however, we need a consensus genetic diagnostic algorithm that integrates different molecular and cytogenetic techniques aiming to collect key leukemogenic events and to carry out a reliable genetic classification (Figure 7A). The challenge is that this rare leukemia subtype exhibits a wide range of complexity and inter-individual variability, thus identifying prognostic and predictive markers, a long-lasting process in large cohorts of prospectively enrolled patients. Furthermore, as each individual case shows various actionable genetic and epigenetic targets [137], a comprehensive study integrating bio-molecular data with drug sensitivity/resistance profile, would be a valuable approach to identify effective combinatorial treatments, thus implementing a personalized medicine (Figure 7B). These new therapeutic opportunities, associated with emerging immunotherapy, i.e., monoclonal antibodies (anti-CD33 in immature T-ALL or anti-CD30 in antigen-expressing cases), or CART-T or NK cell approaches directed against CD4, CD3, CD1a, CD5 T-cell antigens, will hopefully change the outlook of this aggressive subtype of leukemias.

## Figures and Tables

**Figure 1 genes-12-01118-f001:**
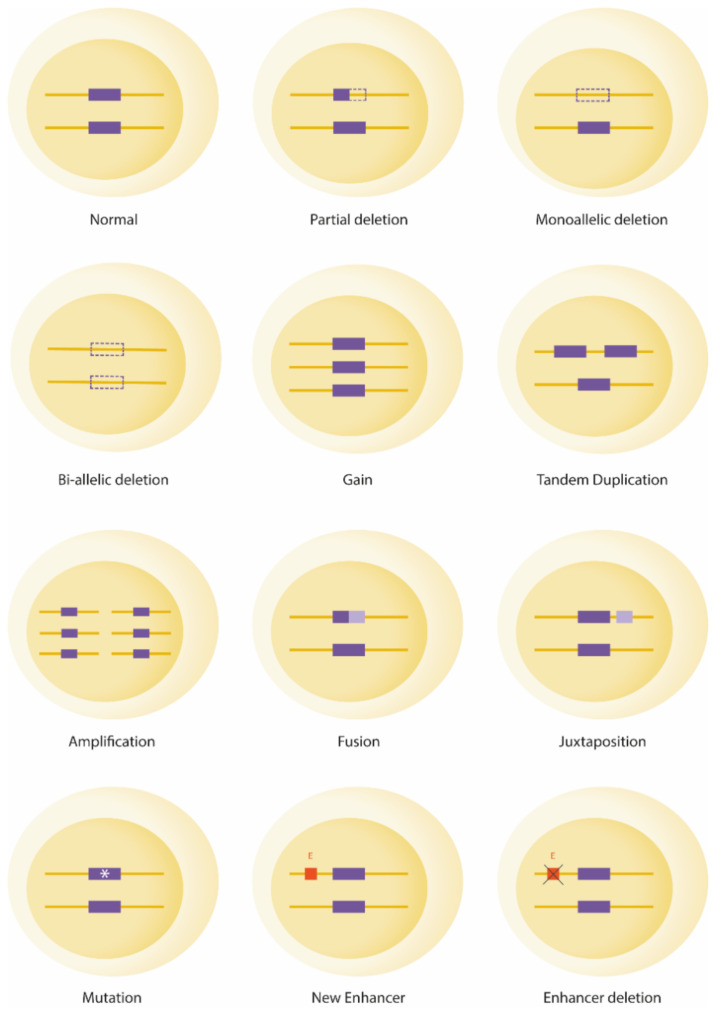
Schematic representation of genomic mechanisms of gene deregulation in T-ALL. Dotted lines: deletion; *: mutation; E: enhancer.

**Figure 2 genes-12-01118-f002:**
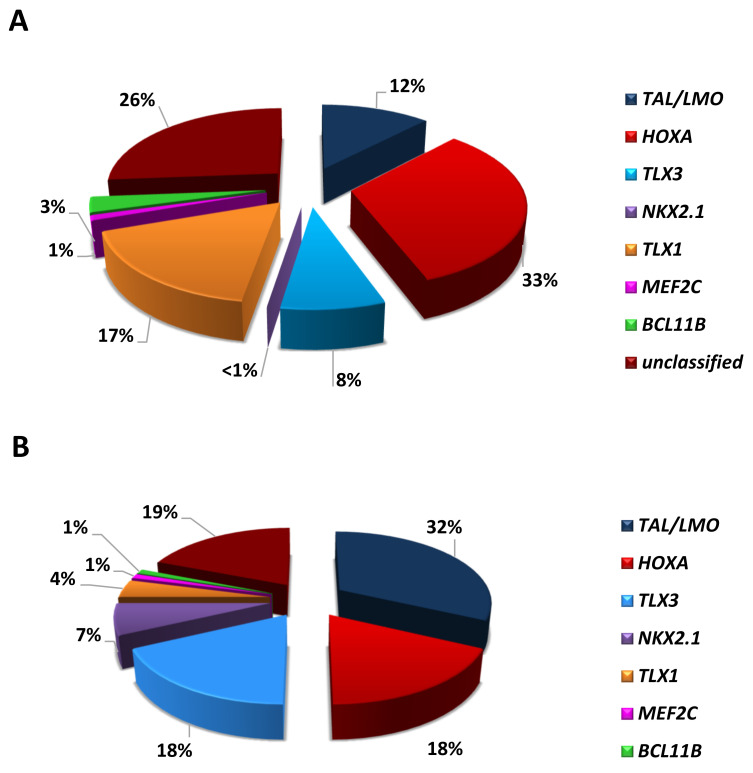
Distribution of the main genetic subgroups in T-ALL of: (**A**) adults and (**B**) children [7,8,9,10].

**Figure 3 genes-12-01118-f003:**
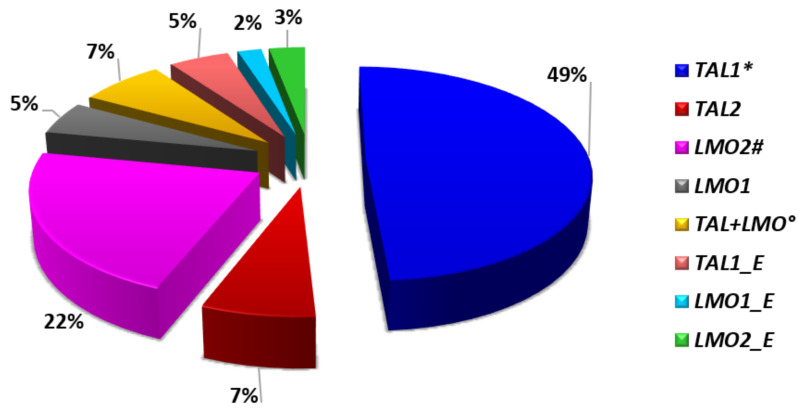
*TAL/LMO* genetic subgroup: Frequency of *TAL* and/or *LMO* rearrangements [8,9]. * In this group, including all rearrangements of *TAL1*, 47% of cases are represented by *STIL-TAL1* fusions; °*TAL1*+*LMO1*, or *TAL1*+*LMO2*, or *TAL1*-*LMO3*; ^#^*LMO2* abnormalities are rarely combined with the rearrangement of *LYL1*; _E, super-enhancer created by mutations of noncoding intergenic sequences [8].

**Figure 4 genes-12-01118-f004:**
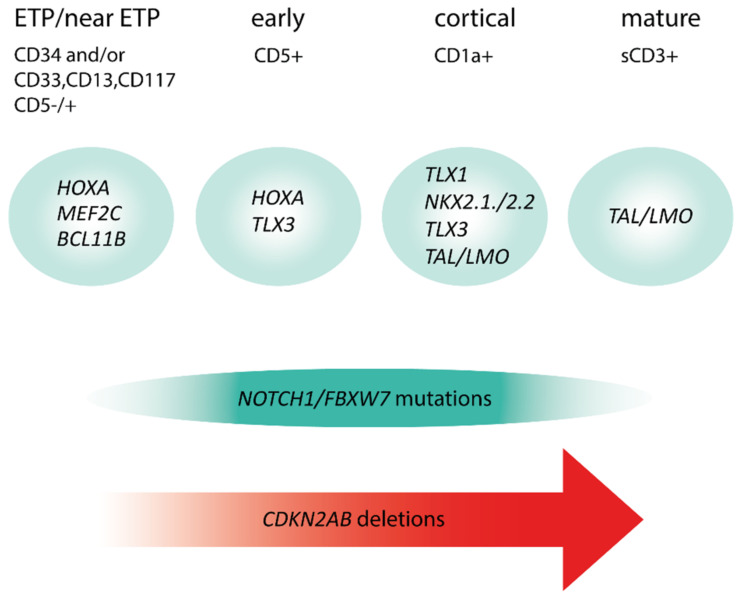
Stages of T-ALL differentiation and distribution of *NOTCH1/FBXW7* mutations and *CDKN2AB* deletions.

**Figure 5 genes-12-01118-f005:**
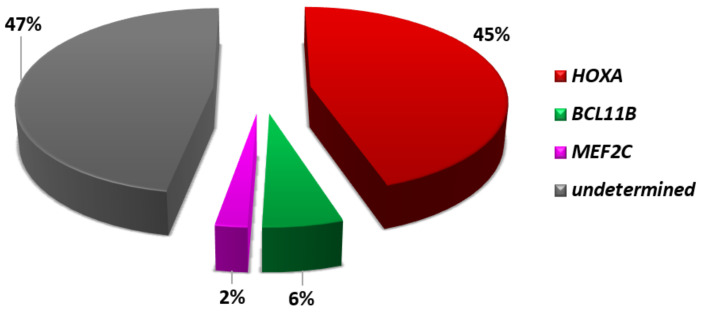
Genetic classification of ETP-ALL [7,8,10].

**Figure 6 genes-12-01118-f006:**
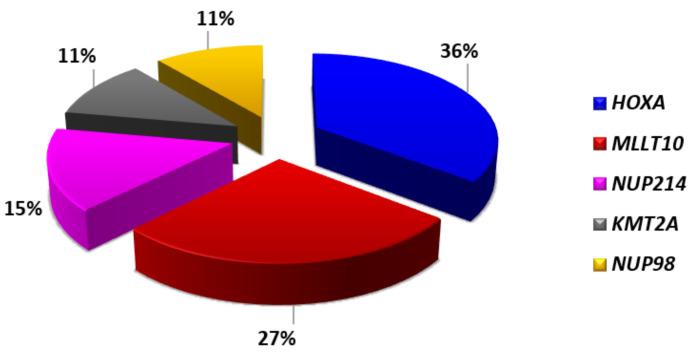
*HOXA* genetic subgroup: Genes associated with high levels of *HOXA* expression and frequency of their alterations [8,9].

**Figure 7 genes-12-01118-f007:**
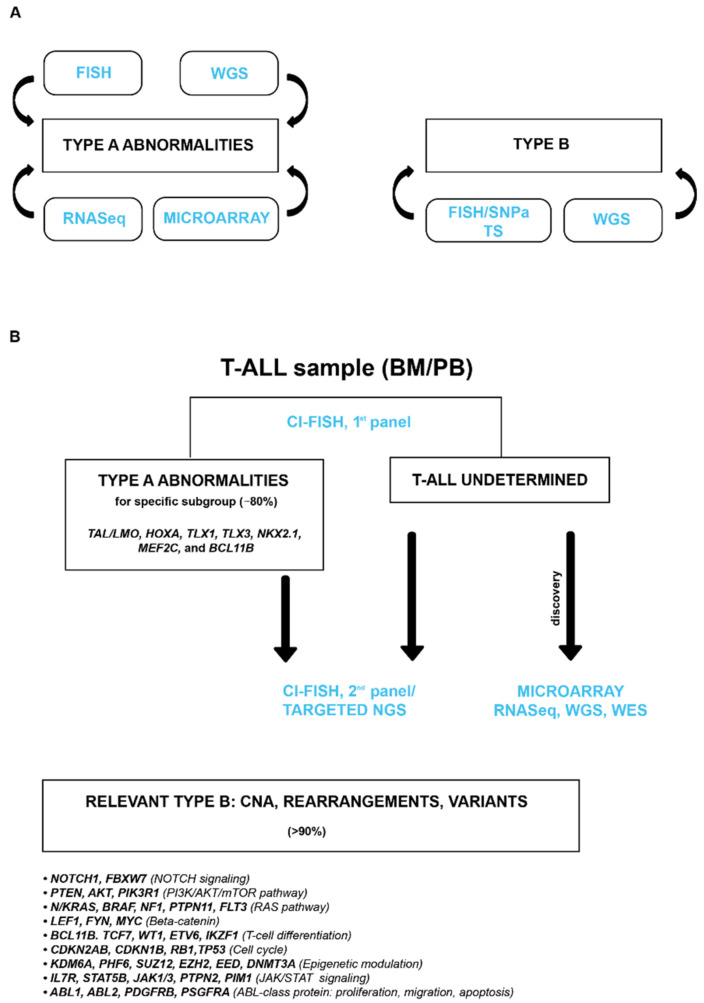
(**A**) Available molecular-cytogenetics techniques; (**B**) Diagnostic algorythm_Personal Experience. CI-FISH (combined interphase-fluorescence in situ hybridization) assay is fully described in La Starza R et al. [8]; CNA, Copy Number Alterations; NGS, Next Generations Sequencing; WGS, Whole Genome Sequencing; WES, Whole Exome Sequencing; RNASeq, RNA sequencing; TS, Targeted Sequencing.

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
