# Peer review of "T-Cell Acute Lymphoblastic Leukemia: Biomarkers and Their Clinical Usefulness"

_genes, 2021, doi:10.3390/genes12081118_

Round 1
Reviewer 1 Report
The review from Bardelli and colleagues, “T-cell Acute Lymphoblastic Leukemia: Biomarkers and Their Clinical Usefulness”, is focused on the recurrent abnormalities that define specific (genetic & epigenetic) leukemogenic pathways in T-cell ALL and that may also serve as diagnostic biomarkers.
This is a comprehensive and generally well written review but needs to be made much more concise and updated. There are also a number of recent extensive reviews (Girardi et al 2017, Cordo’ et al 2021) that similarly describe the oncogenic drivers in the subtypes of T-ALL.
To make this submission more novel and more timely, the authors should perhaps incorporate a comparison to infant T-ALL which appears to be a distinct entity compared to childhood T-ALL (Dorrenberg et al, Mansur et al), and reveals both similar and different genetics with very different outcomes.
A novelty in this manuscript is the authors own BCL11B-a story that could be mentioned in the abstract, and could be reviewed in a little more detail - instead of the lengthy description of the already known players in Section 5. (I’m also not sure why just BCL11b gets a mention in Section 4, and not for example, HOXA or MEF2C. Indeed, perhaps Section 4 should just contain the introduction to ETP, which anyway should come before that of cortical T-ALL).
The section on therapeutic opportunities could also be expanded to include more recent ideas on immune therapies used in T-ALL, micro-RNA etc? What TKI have been used to target BCL11b-a cases of T-ALL?
Minor:
I think Figure 1 is completely unnecessary for this review and the expected readership, the explanation given in the text is sufficient.
Line 49: distinguished as type A and type B
Line 53: allowing the molecular classification of 80-90% of T-ALL
Line 66: characterizes 30-440% of pediatric
Author Response
We would like to thank the Reviewer for his valuable comment and suggestions, which have now been addressed in our revised manuscript.
We have addressed the recommendation to include a paragraph on infant T-ALL which is emerging as a distinct entity from both pediatric and adult T-ALL. From the available studies we have tried to summarize the most relevant knowledge on this subtype of leukemia
As suggested by the Reviewer, the BCL11B-a story was revised and described with more details. We have also included the new findings published by Montefiori LE et al. (Cancer Discovery 2021). Moreover, as suggested, we moved ETP to Section 4 which now comes before cortical T-ALL (Section 5). With this reorganization, BCL11B, HOXA and MEF2C were first mentioned in Section 4.
The Section on therapeutic opportunities has been expanded to include more recent tailored treatements and immune therapy; TKi used to target BCL11B-a acute leukemia were added in Section 4.3.
Minor:
- Figure 1. We agree with the Reviewer that Figure 1 largely replicate the explanation which has been given in the text. However, we are aware that often figures render better than a written explanation the information ones want to discuss, due to the support of the visual representation. For this reason we would like to keep Figure 1, reducing the descriptive part on molecular mechanisms, in the text (see the revised Introduction (Section 1)
- Corrections at lines 49, 53, and 66, were done
Reviewer 2 Report
The review article by Bardelli et al., focuses on recurrent genetic alterations in T-ALL, which define specific disease entities and the leukemogenic pathways involved in disease development, progression and drug resistance as well as on oncogenes and tumorsuppressors that may serve as diagnostic biomarkers.
This is an – at least for the most part – well written article, which provides a concise overview of the current knowledge in the field. However, although I am not a native English speaker myself, I have the feeling that the phrasing/wording in some passages might be improved, that the usage of singular/plural in the corresponding verbs is in some instances incorrect; and the manuscript contains a significant number of typos. I am pointing out only a fraction of these issues in my specific comments and I would strongly recommend that the entire manuscript is proofread by a native speaker.
Specific comments
As the authors also summarize innovative/targeted therapeutic options, this should be briefly mentioned in the abstract to make it even more attractive.
It would add additional value to the manuscript if the authors would give an outlook how they envision a genetic diagnostic algorithm for T-ALL, facilitating the detection of the most relevant genetic alterations. The current obstacle in assessing options for targeted/personalized therapies in T-ALL is – as pointed out by the authors – undeniably their genetic complexity requiring extensive genetic testing, which for many study centers is not feasible in a diagnostic setting.
In all pie charts (Figures 2-3, 5-6) it is unclear from which data sets they are derived; and honestly, they are of rather poor quality and resolution, there are better ways to draw such charts.
The controversial data regarding the prognostic value of TLX3-rearranged T-ALL, in particular in children, should be added.
Not only KDM6A but also PHF6 mutations have been suggested to skew the gender distribution in T-ALL. This should also be mentioned.
I am missing the NOTCH signaling - SKP2 axis, which has emerged as an additional therapeutic vulnerability.
Line 69: “These genetic events countersign distinct subgroups…“ this should be rephrased; “countersign” rather means to countersign a document; the same wording is used e.g. in line 245.
Lines 69-71: “… TAL/LMO, HOXA, 69 TLX3, TLX1, NKX2-1/2-2, MEF2C, allowing the molecular classification 80-90% of T-ALL cases with different distribution in children and adults (Figure 2).” In Figures 2A and 2B, 33% and 18% of cases, respectively, are indicated as unclassified, which is not matching the information provided in the text.
Lines 74-75: “They widespread across all genetic subgroups, even if they do not occur randomly, showing privileged association with the primary genetic change.“ This sentence should be rephrased; at least in my opinion „widespread“ and „privileged association“ are not properly used; also in line 98 the same wording is used.
Line 136: “… DNA duplication …” the more proper phrasing is probably “… DNA replication …”
Lines 139-140: “… CDKN2AB and/or CDKN1B cell cycle regulators, detectable in 80-90% of cases …“ Are TLX1 rearrangements indeed associated with CDKN1B deletions? If so, the incidence of CDKN1B deletions and respective references should be provided.
Lines 274-275: “Nevertheless, the activation of the NOTCH signaling is largely dependent from gain-of-function mutations, that typically occur in two hot-spot domains encoded by exons 27, 28, and 34.“ The affected protein domains should be described, because they are more important than the exon numbers.
All gene names should be double-checked and only those approved by the HGNC (https://www.genenames.org/) should be used; e.g. the approved gene name of SIL is STIL; for TRB@ it is TRB; TR@ and non-TR@ are no longer used, etc.
Several references are not properly formatted.
Author Response
As indicated by the Reviewer, English revision of the manuscript was done
Specific Comments
The availability of innovative/targeted therapeutic options to treat patients with T-ALL was mentioned in the Abstract
A proposal for a diagnostic algorithm combining the genomic characterization of the leukemia together with the possibility to carry out ex-vivo drug sensitivity/resistant profile as been provided as Figure 7.
Figures 2-3, 5-6. References to refer to for the distribution of the major genetic subgroups, in children and adults were provided in the legends. The resolution of all figures was improved.
The conflicting data about the prognostic value of TLX3 in pediatric T-ALL was discussed.
Mutations of both PHF6 and KDM6A were discussed to be probabily linked to skew the gender distribution of T-ALL (see Section 6.6).
Targeting the NOTCH signaling-SKP2 axis was discussed as an additional therapeutic opportunity in NOTCH-dependent T-ALL.
Lines 69, 69-71, 74-75, 136, 139-140, 274-275 were corrected accordingly
All gene names were checked and uniformed according to approved HGNC database
References were re-formatted
Reviewer 3 Report
Bardelli and co-workers provide an extensive yet comprehensive review about the manifold genetic alterations that can be found in the various differentiated subsets of T-ALL. They subdivide their description into logical chapters. First they report how particular types of genetic alterations are related to distinct T-ALL differentiation stages. Then, they discuss how these alterations interact with each other to deregulate specific signaling pathways and, consequently, how they functionally impact various cellular processes that are responsible for leukemia development and progress. Finally, they offer ideas, how these changes might eventually be exploited for novel treatment approaches.
Although the authors obviously tried to give heir very best and despite the fact that this review clearly reflects their deep knowledge and long-standing occupation with this subject, it is still a bit difficult not to lose the oversight and to somehow connect the many facts into a coherent picture. However, as the authors anyway admitted in their last paragraph, they are themselves also well aware of this problem, which of course is definitely not (only) entirely their fault.
There are a few careless mistakes and typos that need to be corrected (e.g. gene names in the abstract need to be italicized; line 83: 440%; line 200 estremely; line 234 this cases)
Author Response
We thank the reviewer for his timely comments and for having grasped the complexity of being able to integrate the numerous and heterogeneous information on the biomolecular characteristics of T-ALL. However, we are convinced that only by transferring this knowledge from the laboratory to the clinic it will be possible to accurately evaluate their clinical significance, i.e. diagnostic, prognostic and / or predictive. This is achievable with prospective multicentre studies, which hopefully will make possible a "precision medicine" for a leukemia in which the therapeutic results obtained to date are still largely unsatisfactory.
Mistakes and typos were corrected throughout the manuscript
Round 2
Reviewer 1 Report
I am happy with all the changes/additions made to this interesting review